# LMSM: A modular approach for identifying lncRNA related miRNA sponge modules in breast cancer

Junpeng Zhang[1,2‡*], Taosheng Xu[3‡], Lin Liu[4], Wu Zhang[5], Chunwen Zhao[2], Sijing Li[2], Jiuyong Li[4], Nini Rao[1*], Thuc Duy Le[4*]

1 Center for Informational Biology, School of Life Science and Technology, University of Electronic Science and Technology of China, Chengdu, Sichuan, China, 2 School of Engineering, Dali University, Dali, Yunnan, China, 3 Institute of Intelligent Machines, Hefei Institutes of Physical Science, Chinese Academy of Sciences, Hefei, Anhui, China, 4 School of Information Technology and Mathematical Sciences, University of South Australia, Mawson Lakes, SA, Australia, 5 School of Agriculture and Biological Sciences, Dali University, Dali, Yunnan, China

‡ These authors share first authorship of this work.
* zhangjunpeng_411@yahoo.com (JZ); raonn@uestc.edu.cn (NR); thuc.le@unisa.edu.au (TDL)

**Data Availability Statement:** All relevant data are within the manuscript and its Supporting Information files.

## Abstract

Until now, existing methods for identifying lncRNA related miRNA sponge modules mainly rely on lncRNA related miRNA sponge interaction networks, which may not provide a full picture of miRNA sponging activities in biological conditions. Hence there is a strong need of new computational methods to identify lncRNA related miRNA sponge modules. In this work, we propose a framework, LMSM, to identify LncRNA related MiRNA Sponge Modules from heterogeneous data. To understand the miRNA sponging activities in biological conditions, LMSM uses gene expression data to evaluate the influence of the shared miRNAs on the clustered sponge lncRNAs and mRNAs. We have applied LMSM to the human breast cancer (BRCA) dataset from The Cancer Genome Atlas (TCGA). As a result, we have found that the majority of LMSM modules are significantly implicated in BRCA and most of them are BRCA subtype-specific. Most of the mediating miRNAs act as crosslinks across different LMSM modules, and all of LMSM modules are statistically significant. Multi-label classification analysis shows that the performance of LMSM modules is significantly higher than baseline's performance, indicating the biological meanings of LMSM modules in classifying BRCA subtypes. The consistent results suggest that LMSM is robust in identifying lncRNA related miRNA sponge modules. Moreover, LMSM can be used to predict miRNA targets. Finally, LMSM outperforms a graph clustering-based strategy in identifying BRCA-related modules. Altogether, our study shows that LMSM is a promising method to investigate modular regulatory mechanism of sponge lncRNAs from heterogeneous data.

## Author summary

Previous studies have revealed that long non-coding RNAs (lncRNAs), as microRNA (miRNA) sponges or competing endogenous RNAs (ceRNAs), can regulate the expression

**Funding:** JZ was supported by the National Natural Science Foundation of China (Grant Number: 61702069, 61963001), the Applied Basic Research Foundation of Science and Technology of Yunnan Province (Grant Number: 2017FB099). LL and JL were supported by the Australian Research Council Discovery Grant (Grant Number: DP170101306). TX was supported by the National Natural Science Foundation of China (Grant Number: 61902372). WZ was supported by the Education Science Research Foundation of Yunnan Province (Grant Number: 2018JS416). NR was supported by the National Natural Science Foundation of China (Grant Number: 61872405, 61720106004). TDL was supported by NHMRC Grant (Grant Number: 1123042). The funders had no role in study design, data collection and analysis, decision to publish, or preparation of the manuscript.

**Competing interests:** The authors have declared that no competing interests exist.

levels of messenger RNAs (mRNAs) by decreasing the amount of miRNAs interacting with mRNAs. In this work, we hypothesize that the "tug-of-war" between RNA transcripts for attracting miRNAs is across groups or modules. Based on the hypothesis, we propose a framework called LMSM, to identify LncRNA related MiRNA Sponge Modules. Based on the two miRNA sponge modular competition principles, significant sharing of miRNAs and high canonical correlation between the sponge lncRNAs and mRNAs, LMSM is also capable of predicting miRNA targets. LMSM not only extends the ceRNA hypothesis, but also provides a novel way to investigate the biological functions and modular mechanism of lncRNAs in breast cancer.

## Introduction

Long non-coding RNAs (lncRNAs) are RNA transcripts with more than 200 nucleotides (nts) in length [1]. More and more evidence has shown that lncRNAs play important functional roles in many biological processes, including human cancers [2–4]. As a major class of non-coding RNAs (ncRNAs), lncRNAs have attracted increasing interest from researchers in their exploration of non-coding knowledge from the 'junk'.

Among the wide range of biological functions of lncRNAs, their role as competing endogenous RNAs (ceRNAs) or miRNA sponges is in the limelight. As a family of small ncRNAs (~18nts in length), miRNAs are important post-transcriptional regulators of gene expression [5,6]. According to the ceRNA hypothesis [7], lncRNAs contain abundant miRNA response elements (MREs) for competitively sequestering target mRNAs from miRNAs' control. This regulation mechanism of lncRNAs when acting as miRNA sponges is highly implicated in various human diseases [8], including breast cancer [9]. For example, lncRNA *H19*, an imprinted gene is associated with breast cancer cell clonogenicity, migration and mammosphere-forming ability. By sponging miRNA *let-7*, *H19* forms a *H19/let-7/LIN28* reciprocal negative regulatory circuit to play a critical role in the breast cancer stem cell maintenance [10].

To systematically investigate the functions of lncRNAs as miRNA sponges in human cancer, a series of computational methods have been developed to infer lncRNA related miRNA sponge interaction networks. The methods can be divided into three categories according to the statistical or computational techniques employed: pair-wise correlation based approach, partial association based approach, and mathematical modelling approach [11].

It is commonly known that to implement a specific biological function, genes tend to cluster or connect in the form of modules or communities. Consequently, based on the identified lncRNA related miRNA sponge interaction networks, several methods [12–17] using graph clustering algorithms were developed to identify lncRNA related miRNA sponge modules. For the identification of sponge lncRNA-mRNA pairs, most of existing methods only consider pair-wise correlation of them. Since the lncRNA related miRNA sponge interaction networks are created by simply putting together sponge lncRNA-mRNA pairs, when the expression levels of each sponge lncRNA-mRNA pair are highly correlated, the collective correlation between the set of sponge lncRNAs and the set of mRNAs in the same identified module is not necessarily high. As we know, the pair-wise positive correlation between the expression levels of a lncRNA and a mRNA pair is commonly used to identify the sponge interactions between them. For the identification of lncRNA related miRNA sponge modules, it is also necessary to investigate whether the clustered sponge lncRNAs and mRNAs in a module have high collective positive correlation or not. Moreover, these methods do not consider the influence of the shared miRNAs on the expression of the clustered sponge lncRNAs and mRNAs. It is known

that the "tug-of-war" between sponge lncRNAs and mRNAs is mediated by miRNAs. Therefore, it is extremely important to consider the influence of the shared miRNAs in identifying lncRNA related miRNA sponge modules.

Recently, to study lncRNA, miRNA and mRNA-associated regulatory modules, Deng *et al.* [18] and Xiao *et al.* [19] have proposed two types of joint matrix factorization methods to identify mRNA-miRNA-lncRNA co-modules by integrating gene expression data and putative miRNA-target interactions. However, it is still not clear how the shared miRNAs influence the expression level of the sponge lncRNAs and mRNAs in a module.

To address the above issues, we firstly hypothesize that sponge lncRNAs form a group to competitively release a group of target mRNAs from the control of the miRNAs shared by the lncRNAs and mRNAs (details see Section Materials and methods). We name this hypothesis the *miRNA sponge modular competition hypothesis* in this paper. Then based on the hypothesis, we propose a novel framework to identify LncRNA related MiRNA Sponge Modules (LMSM). The framework firstly uses the WeiGhted Correlation Network Analysis (WGCNA) [20] method to generate lncRNA-mRNA co-expression modules. Next, by incorporating matched miRNA expression and putative miRNA-target interactions, LMSM applies three constraints (see Section Materials and methods) to obtain lncRNA related miRNA sponge modules (also called LMSM modules in this paper). One of the constraints, high canonical correlation, is used to assess whether the group of sponge lncRNAs and the group of mRNAs in the same module have a high collective positive correlation or not. The other constraint, adequate sensitivity canonical correlation conditioning on a group of miRNAs, is used to evaluate the influence of the shared miRNAs on the clustered sponge lncRNAs and mRNAs.

To evaluate the LMSM approach, we apply it to matched miRNA, lncRNA and mRNA expression data, and clinical information of breast cancer (BRCA) dataset from The Cancer Genome Atlas (TCGA). The modular analysis results demonstrate that LMSM can help to uncover modular regulatory mechanism of sponge lncRNAs in BRCA. LMSM is released under the GPL-3.0 License, and is freely available through GitHub repository (https://github.com/zhangjunpeng411/LMSM).

## Materials and methods

### A hypothesis on miRNA sponge modular competition

The ceRNA hypothesis [7] indicates that a pool of RNA transcripts (known as ceRNAs) regulate each other's transcripts by competing for the shared miRNAs through MREs. Based on this unifying hypothesis, a large-scale gene regulatory network including coding and non-coding RNAs across the transcriptome can be formed, and it plays critical roles in human physiological and pathological processes. However, by using MREs as letters of language, the hypothesis only depicts the crosstalk between individual RNA transcript (e.g. coding RNAs, lncRNAs, circRNAs or pseudogenes) and mRNA at the pair-wise interaction level and the crosstalk between RNA transcripts and mRNAs at the module level is still an open question.

There has been evidence showing that for the same transcriptional regulatory program, biological process or signaling pathway, genes tend to form modules or communities to coordinate biological functions [21]. These modules correspond to functional units in complex biological systems, and they play important role in gene regulation. Based on these findings, in this paper, we hypothesize that regarding miRNA sponging, the crosstalk between different RNA transcripts is in the form of modular competition. We call the hypothesis the *miRNA sponge modular competition hypothesis*.

As shown in Fig 1, based on our hypothesis, instead of having pair-wise competitions, miRNA sponges form groups to compete at module level for common miRNAs. Here, a

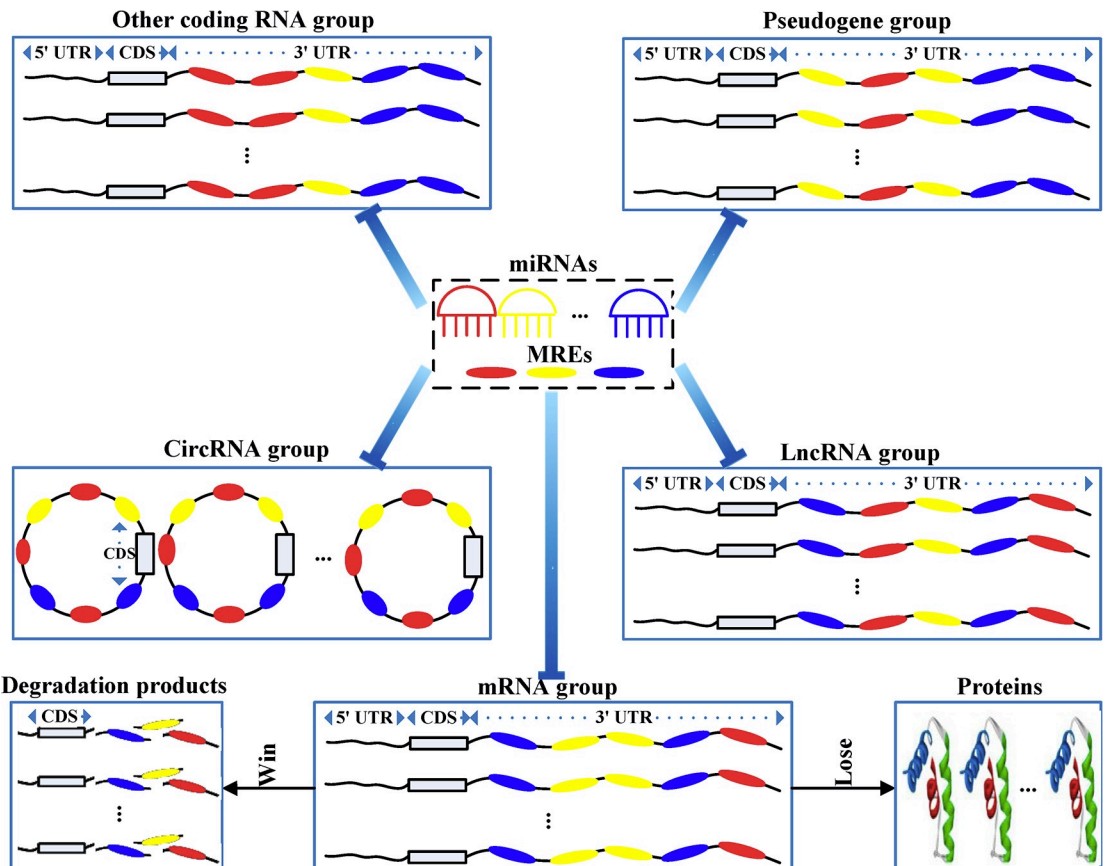

**Fig 1. An illustration of the miRNA sponge modular competition hypothesis.** The four types of miRNA sponges (other coding RNAs, lncRNAs, circRNAs or pseudogenes), miRNAs and their target mRNAs are shown. Each miRNA sponge module consists of a group of the same type of miRNA sponges, e.g. a group of lncRNAs and a group of target mRNAs. In the same module, the group of miRNA sponges competes with the group of target mRNAs for binding with a set of miRNAs. If the miRNA sponges win the competition, the group of target mRNAs will be released from repression and they will be translated into proteins. If the miRNA sponges lose the competition, the group of target mRNAs will be post-transcriptionally repressed and degraded.

miRNA sponge module consists of a competing group (other coding RNA group, pseudogene group, circRNA group or lncRNA group) and a mRNA group. Here, other coding RNAs also include mRNAs. From the perspective of modularity, the hypothesis at module level extends the ceRNA hypothesis and provides a new channel to look into the functions and regulatory mechanism of miRNA sponges or ceRNAs. Since the available resources of lncRNAs are more abundant than those of other non-coding RNAs (e.g. circRNAs and pseudogenes), in this paper, we focus on the competition between lncRNAs and mRNAs to validate and demonstrate the proposed *miRNA sponge modular competition hypothesis*. Our goal is to discover lncRNA related sponge modules, or LMSM modules. Here each LMSM module contains a group of lncRNAs which compete collectively with a group of mRNAs for sponging the same set of miRNAs.

## The LMSM framework

**Overview of LMSM.** As shown in Fig 2, the proposed LMSM framework comprises two stages. In stage 1, the WGCNA method [20] is used for finding lncRNA-mRNA co-expression modules from matched lncRNA and mRNA expression data. Then in stage 2, LMSM identifies

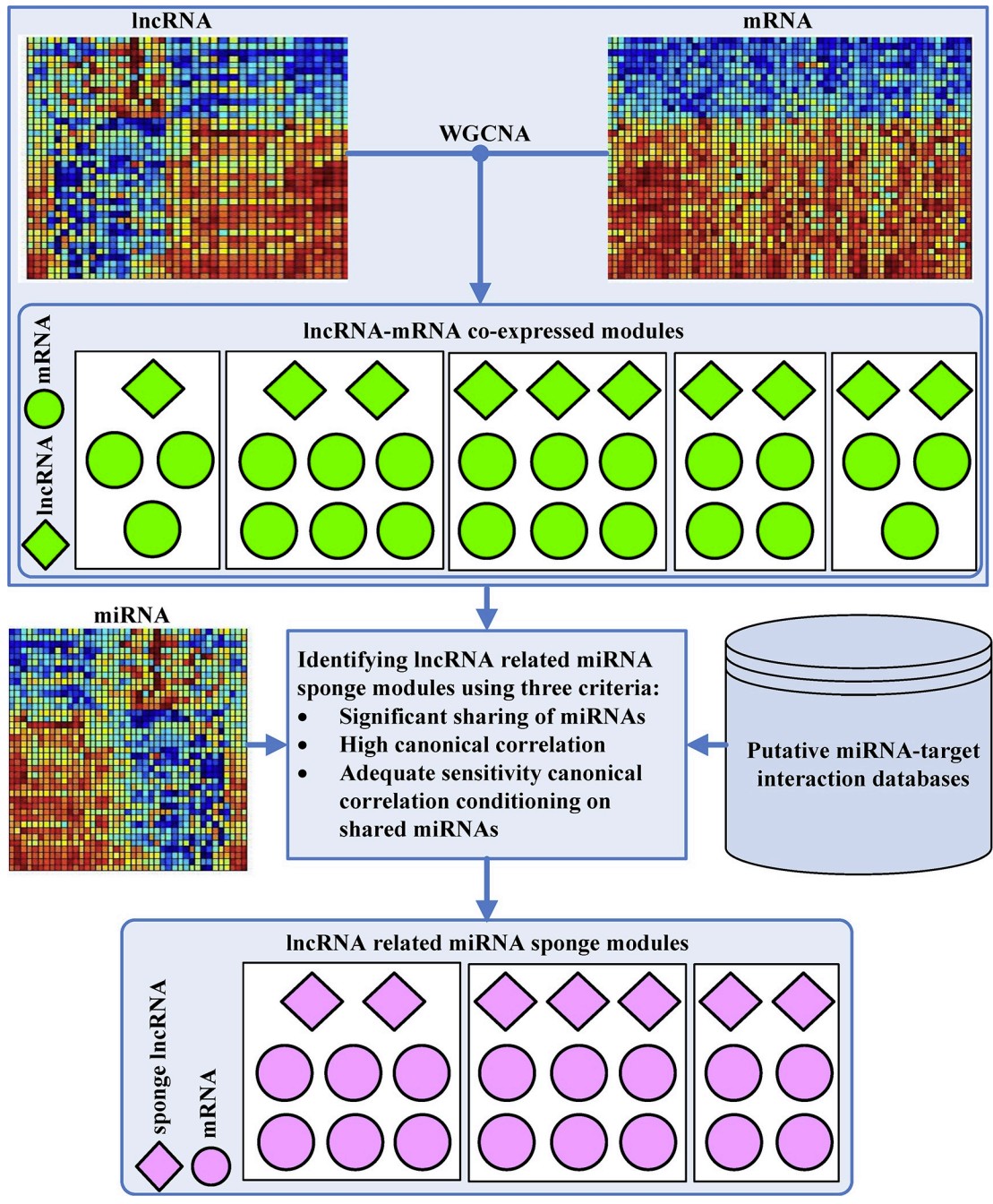

**Fig 2. Workflow of LMSM.** Firstly, we use the WGCNA method to infer lncRNA-mRNA co-expression modules from the matched lncRNA and mRNA expression. Then by using miRNA expression data and putative miRNA-target interactions, we infer lncRNA related miRNA sponge modules (LMSM) by applying three criteria: significant sharing of miRNAs by the group of lncRNAs and the group of target mRNAs in the same co-expression module, high canonical correlation between the lncRNA group and the target mRNA group, and adequate sensitivity canonical correlation between the lncRNA group and the target mRNA group conditioning on shared miRNAs. Each LMSM module must contain at least two sponge lncRNAs and two target mRNAs.

LMSM modules from the lncRNA-mRNA co-expression modules using three criteria. That is, a co-expression module is considered as a LMSM module if the group of lncRNAs and the

group of mRNAs in the co-expression module: (1) have significant sharing of miRNAs, (2) have high canonical correlation between their expression levels, and (3) have adequate sensitivity canonical correlation conditioning on their shared miRNAs. LMSM checks the criteria one by one, and once a co-expression module does not meet a criterion, it is discarded and will not be checked for the next criterion. In the following, we will describe the two stages in detail.

**Identifying lncRNA-mRNA co-expression modules.** For identifying lncRNA-mRNA co-expression modules, we use the WGCNA method. WGCNA is a popular method for identifying co-expressed genes across samples and it can be used to identify clusters of highly co-expressed lncRNAs and mRNAs. In our task, we use the matched lncRNA and mRNA expression data as input to the *WGCNA* R package [20] to identify lncRNA-mRNA co-expression modules. We use the scale-free topology criterion for soft thresholding. The coefficient of determination $R^2$ (the range is from 0 to 1) is used to quantify the goodness of scale-free topology, and larger $R^2$ values mean better scale-free topology. Normally, the $R^2$ value larger than 0.8 in power law curve fit is ranked as good-level in the WGCNA method. Therefore, the desired minimum scale free topology fitting index $R^2$ is set as 0.8 in this work.

**Inferring lncRNA related miRNA sponge modules.** To identify lncRNA related miRNA sponge modules from the co-expression modules obtained in stage 1, we propose three criteria (detailed below) by following the key tenet of our *miRNA sponge modular competition hypothesis*. That is, a group of lncRNAs (acting as miRNA sponges) competes with a group of mRNAs with respect to a set of miRNAs shared by the two groups.

The first criterion requires that the group of lncRNAs and the group of mRNAs in a miRNA sponge module have a significant sharing of a set of miRNAs. LMSM uses a hypergeometric test to assess the significance of the sharing of miRNAs between the group of lncRNAs and the group of mRNAs in a co-expression module, based on putative miRNA-target interactions. The *p*-value for the test is computed as:

$$p-value = 1 - \sum_{i_1=0}^{L_1-1} \frac{\binom{M_1}{i_1}\binom{N_1-M_1}{K_1-i_1}}{\binom{N_1}{K_1}} \tag{1}$$

In the equation, $N_1$ is the number of all miRNAs in the dataset, $M_1$ and $K_1$ denote the total numbers of miRNAs interacting with the group of lncRNAs and the group of mRNAs in the co-expression module respectively, and $L_1$ (e.g. 3) is the number of miRNAs shared by the group of lncRNAs and the group of mRNAs in the co-expression module.

The second criterion is to assure that the sponge modular competition between the group of lncRNAs and the group of mRNAs in a miRNA sponge module is strong enough. In existing work, to identify lncRNA related mRNA sponge interactions, a principle followed is that the expression level of a lncRNA and the expression level of a mRNA need to be strongly and positively correlated. Following the same principle on strong positive correlation in expression levels while considering our modular competition hypothesis, LMSM requires the *collective* correlation between the expression levels of the group of lncRNAs and the group of target mRNAs in the same module to be strong and positive. To assess the *collective* correlation, we perform canonical correlation analysis [22] to obtain the canonical correlation between the group of lncRNAs and the group of mRNAs in a co-expression module. Let the two column vectors $X = (x_1, x_2, \ldots, x_m)^T$ and $Y = (y_1, y_2, \ldots, y_n)^T$ represent the group of lncRNAs and the group of mRNAs in a co-expression module respectively. $\Sigma_{XX}$, $\Sigma_{YY}$ and $\Sigma_{XY}$ are the variance or cross-covariance matrices calculated from the expression data of $X$ and $Y$. The canonical

correlation analysis seeks the canonical vectors $a$ ($a \in \mathbb{R}^m$) and $b$ ($b \in \mathbb{R}^n$) which maximize the correlation of $\text{corr}(a^T X, b^T Y)$. The canonical correlation between the group of lncRNAs and the group of mRNAs, denoted as $CC_{lncR-mR}$, is then calculated as follows with the found canonical vectors:

$$CC_{lncR-mR} = \text{corr}(a^T X, b^T Y) = \frac{a^T \sum_{XY} b}{\sqrt{a^T \sum_{XX} a} \sqrt{b^T \sum_{YY} b}} \qquad 2$$

In this work, we use the *PMA* R package [23] to compute canonical correlation.

Finally, the third criterion adapted from the sensitivity correlation [24] is employed to assess if the miRNAs shared by the group lncRNAs and the group of mRNAs in a module have large enough influence on the modular competition between the two groups of RNAs. To check according to this criterion, we incorporate miRNA expression data, and compute $SCC_{lncR-mR}$ the sensitivity canonical correlation between the group of lncRNAs and the group of mRNAs in a co-expression module as follows:

$$SCC_{lncR-mR} = CC_{lncR-mR} - PCC_{lncR-mR} \qquad 3$$

where $PCC_{lncR-mR}$ is the partial canonical correlation between the group of lncRNAs and the group of mRNAs, i.e. the canonical correlation conditioning on the expression of their shared miRNAs in the co-expression module, or the canonical correlation between the two groups of RNAs when the influence of the shared miRNAs is eliminated. Therefore, from Eq (3), we see that $SCC_{lncR-mR}$ implies the correlation between the two groups of RNAs under the influence of their shared miRNAs.

$PCC_{lncR-mR}$ in Eq (4) can be calculated as:

$$PCC_{lncR-mR} = \frac{CC_{lncR-mR} - CC_{miR-mR} CC_{miR-lncR}}{\sqrt{1 - CC_{miR-mR}^2} \sqrt{1 - CC_{miR-lncR}^2}} \qquad 4$$

where $CC_{miR-mR}$ ($CC_{miR-lncR}$) is the canonical correlation between the set of miRNAs in the co-expression module and the group of mRNAs (lncRNAs) in the co-expression module.

In this study, empirically, a lncRNA-mRNA co-expressed module with *p*-value $< 0.05$ for the hypergeometric test of miRNA sharing (criterion 1), $CC_{lncR-mR} > 0.8$ for modular competition strength assessment (criterion 2) and $SCC_{lncR-mR} > 0.1$ for miRNA influence (criterion 3) is regarded as a lncRNA related miRNA sponge module (a LMSM module).

## Evaluating statistical significance of LMSM modules

To evaluate the statistical significance of LMSM modules, we adapt the null model method proposed in [25]. The null model method hypothesizes that the shared miRNAs do not affect the correlation between two genes, i.e. the sensitivity correlation (the difference between correlation and partial correlation) between two genes is 0, and has been successfully applied to evaluate statistical significance of ceRNA interactions. Similar to [25], LMSM is also adapted from the Sensitivity Correlation (SC) method [24]. Therefore, the null model method can be applied to evaluate the statistical significance of LMSM modules. In our null model, the null hypothesis is that the group of the shared miRNAs does not influence the canonical correlation between the group of lncRNAs and the group of mRNAs, i.e $SCC_{lncR-mR} = 0$. For each LMSM module, a group of lncRNAs or a group of mRNAs corresponds to a gene, and a group of the shared miRNAs corresponds to a miRNA in the null model. For obtaining more precise *p*-values, the number of datasets sampled is set to 1E+06 for the null model. Since the sampling procedure is computationally intensive, we use the pre-computed sets of covariance matrices in *SPONGE* R package [25] to build our null model. Based on the constructed null model, we can

infer adjusted *p*-values (adjusted by Benjamini and Hochberg method [26]) for each LMSM module. A LMSM module with adjusted *p*-value less than 0.05 is regarded as a statistically significant module.

## Application of LMSM in BRCA

**BRCA enrichment analysis.** Instead of performing Gene Ontology (GO) and Kyoto Encyclopedia of Genes and Genomes Pathway (KEGG) enrichment analysis, to investigate whether an identified LMSM module is functionally associated with BRCA, we focus on conducting BRCA enrichment analysis by using a hypergeometric test. For a LMSM module, the *p*-value for the test is calculated as:

$$p - value = 1 - \sum_{i_2=0}^{L_2-1} \frac{\binom{M_2}{i_2}\binom{N_2 - M_2}{K_2 - i_2}}{\binom{N_2}{K_2}} \qquad 5$$

where $N_2$ is the number of genes (lncRNAs and mRNAs) in the dataset, $M_2$ denotes the number of BRCA genes in the dataset, $K_2$ represents the number of genes in the LMSM module, and $L_2$ is the number of BRCA genes in the LMSM module. A LMSM module with *p*-value < 0.05 is regarded as a BRCA-related module.

**Module biomarker identification in BRCA.** The module survival analysis can imply whether the identified LMSM modules are good biomarkers of the metastasis risks of cancer patients or not, and it can give us the hint whether the LMSM modules may be related to and potentially affect the metastasis or survival of cancer patients. For each BRCA sample, we fit the multivariate Cox model (proportional hazards regression model) [27] using the genes (lncRNAs and mRNAs) in LMSM modules to compute its risk score. All the BRCA samples are equally divided into the high risk and the low risk groups according to their risk scores. The Log-rank test is used to evaluate the difference of each LMSM module between the high and the low risk BRCA groups. Moreover, we also calculate the proportional hazard ratio (HR) between the high and the low risk BRCA groups. In this work, the *survival* R package [28] is utilized, and a LMSM module with Log-rank *p*-value < 0.05 and HR > 2 is regarded as a module biomarker in BRCA.

**Identification of BRCA subtype-specific modules.** As is known, BRCA is a heterogeneous disease with several molecular subtypes, and the choice of chemotherapy for each BRCA subtype is different. This diversity indicates that the genetic regulation of each BRCA subtype is specific. To identify BRCA subtype-specific modules, we firstly identify BRCA molecular subtypes using the PAM50 classifier [29]. By using a 50-gene subtype predictor, the PAM50 classifier classifies a BRCA sample into one of the five "intrinsic" subtypes: Luminal A (LumA), Luminal B (LumB), HER2-enriched (Her2), Basal-like (Basal) or Normal-like (Normal). In this work, we use the *genefu* R package [30] to predict molecular subtypes of each BRCA sample in the dataset used in our study.

To identify BRCA subtype-specific LMSM modules, we firstly need to estimate the enrichment scores of LMSM modules in BRCA samples. To calculate the enrichment score of each LMSM module in BRCA samples, the gene set variation analysis (GSVA) method [31] is used. To calculate the enrichment score, the GSVA method uses the Kolmogorov-Smirnov (KS) like

random walk statistic as follows:

$$v_{jk}(\ell) = \frac{\sum_{i=1}^{\ell} |r_{ij}|^{\tau} I(g(i) \in \gamma_k)}{\sum_{i=1}^{p} |r_{ij}|^{\tau} I(g(i) \in \gamma_k)} - \frac{\sum_{i=1}^{\ell} I(g(i) \notin \gamma_k)}{p - |\gamma_k|} \qquad 6$$

where $\tau(\tau = 1$ by default) is the weight of the tail in the random walk, $r_{ij}$ is the normalized expression-level statistics of the $i$-th gene in the $j$-th sample as defined in [31], $\gamma_k$ is the $k$-th LMSM module, $I(g(i) \in \gamma_k)$ is the indicator function on whether the $i$-th gene belongs to the LMSM module $\gamma_k$, $|\gamma_k|$ is the number of genes in the $k$-th LMSM module, and $p$ is the number of genes in the dataset.

To transform the KS like random walk statistic into an enrichment score (*ES*, also called GSVA score), we calculate the maximum deviation from zero of the random walk of the $j$-th sample with respect to the $k$-th LMSM module in the following:

$$ES_{jk}^{\max} = v_{jk}[\arg \max_{\ell=1,\dots,p} (abs(v_{jk}(\ell)))] \qquad 7$$

For each LMSM module $\gamma_k$, the formula generates a distribution of enrichment scores that is bimodal (see the reference [31] for a more detailed description).

Based on the enrichment scores of LMSM modules in each BRCA sample, we further identify two types of BRCA subtype-specific LMSM modules, up-regulated modules and down-regulated modules. For one type of regulation pattern (up or down regulation), a LMSM module is regarded to be specific to a BRCA subtype. For an up-regulated BRCA subtype-specific LMSM module, the enrichment score of the LMSM module in the specific BRCA subtype samples is significantly larger than the score in the other BRCA subtype samples. For a down-regulated BRCA subtype-specific LMSM module, the enrichment score of the LMSM module in the specific BRCA subtype samples is significantly smaller than the score in the other BRCA subtype samples. For example, if a LMSM module $\gamma_k$ is up-regulated Basal-like specific, the enrichment scores of the LMSM module in Basal-like samples should be significantly larger than those in Luminal A, Luminal B, HER2-enriched and Normal-like samples. In this work, for each LMSM module, we use Welch's *t*-test [32] to calculate the significance *p*-value for the difference of the average enrichment scores between any two BRCA subtype samples. Given a BRCA subtype, a LMSM module is considered as an up-regulated (or down-regulated) module specific to this BRCA subtype if the module's average enrichment score in samples of the given subtype is higher (or smaller) than the average enrichment score in samples of any other subtype and the significance *p*-value of the Welch's *t*-test between the samples of this subtype and any other subtype is less than 0.05.

**Performance of LMSM modules in classifying BRCA subtypes.** In this section, to check the biological relevance of the discovered LMSM modules, we conduct module classification of BRCA subtypes. Here, classifying BRCA subtypes (LumA, LumB, Her2, Basal and Normal) is a multi-class classification (also known as a special case of multi-label classification). To understand the classification performance of the feature genes in each LMSM module, we apply a state-of-the-art multi-label learning strategy called Binary Relevance (BR) [33] implemented in the *utiml* R package [34] to conduct multi-label classification analysis. For the BR strategy, we use the Support Vector Machine (SVM) classifier [35] with default parameters implemented in *e1071* R package [36] as the base algorithm to build the multi-label model. We select two commonly used multi-label classification measures: *Subset accuracy* and *Hamming loss*, and conduct 10-fold cross-validation to evaluate the performance of each LMSM module. In this work, *Subset accuracy* denotes the percentage of correct predictions and *Hamming loss* is the fraction of wrong predictions to the total number of predictions. Higher values of *Subset*

*accuracy* and smaller values of *Hamming loss* indicate better classification performance. In addition, for the evaluation, we use the baseline method in [37], a commonly used multi-label classification method as the baseline for comparison. The base algorithm of the baseline method is also the SVM classifier with default parameters implemented in *e1071* R package [36].

## Results

### Heterogeneous data sources

We collect matched miRNA, lncRNA and mRNA expression data, and clinical data of BRCA dataset from The Cancer Genome Atlas (TCGA, https://cancergenome.nih.gov/). A lncRNA or mRNA without a corresponding gene symbol in the expression data of BRCA dataset is removed. To obtain a unique expression value for replicates of miRNAs, lncRNAs or mRNAs, we compute the average expression value of the replicates. As a result, we obtain the matched expression data of 674 miRNAs, 12711 lncRNAs and 18344 mRNAs in 500 BRCA samples.

We retrieve putative miRNA-target interactions (including miRNA-lncRNA and miRNA-mRNA interactions) from several high-confidence miRNA-target interaction databases and use the combined database search results. Specifically, the putative miRNA-lncRNA interactions are obtained from NPInter v3.0 [38] and the experimental module of DIANA-LncBase v2.0 [39], and miRNA-mRNA interactions are from miRTarBase v8.0 [40], TarBase v7.0 [41] and miRWalk v2.0 [42].

The BRCA related mRNAs are from DisGeNET v5.0 [43] and COSMIC v86 [44], and the BRCA related lncRNAs are from LncRNADisease v2.0 [45], Lnc2Cancer v2.0 [46] and MNDR v2.0 [47]. The ground truth of lncRNA related miRNA sponge interactions is obtained by integrating the interactions from miRSponge [48], LncCeRBase [49] and LncACTdb v2.0 [50].

### Most of the mediating miRNAs act as crosslinks across LMSM modules

Following the steps shown in Fig 2, we have identified 17 LMSM modules (details can be seen in S1 Data). The average size of the identified modules is 672.53 and the average number of the shared miRNAs in a module is 232.82. In total, there are 549 unique miRNAs mediating the 17 LMSM modules, and 90.16% (495 out of 549) miRNAs mediate at least two LMSM modules (details can be seen in S2 Data). This result indicates that most of the mediating miRNAs act as crosslinks across different LMSM modules.

### LMSM modules are all statistically significant

In this section, by computing null-model-based *p*-values, we evaluate whether the identified LMSM modules are statistically significant or not. As a result, the adjusted *p*-values for the identified 17 LMSM modules (details can be seen in S3 Data) are all statistically significant (adjusted *p*-value = 1.00E-06). This result demonstrates that LMSM modules are all statistically significant.

### Most of LMSM modules are implicated in BRCA

To investigate whether the identified LMSM modules are related to BRCA or not, we conduct BRCA enrichment analysis and identify BRCA module biomarkers using the methods described in Section Materials and methods. For the BRCA enrichment analysis, we have collected a list of 4819 BRCA genes (734 BRCA lncRNAs and 4085 BRCA mRNAs) associated with the matched lncRNA and mRNA expression data (details in S4 Data). As shown in Table 1, 10 out of 17 LMSM modules are functionally enriched in BRCA at a significant level

Table 1. **BRCA-related LMSM modules.** $L_2$ is the number of BRCA genes in each LMSM module, $K_2$ represents the number of genes in each LMSM module, the number of BRCA genes in the dataset ($M_2$) is 4819, and the number of genes in the dataset ($N_2$) is 31055.

| Module ID | $L_2$ | $K_2$ | p-value |
|---|---|---|---|
| LMSM 2 | 327 | 1338 | 0 |
| LMSM 3 | 259 | 1340 | 7.34E-05 |
| LMSM 4 | 78 | 392 | 1.14E-02 |
| LMSM 5 | 89 | 449 | 8.07E-03 |
| LMSM 6 | 88 | 370 | 1.97E-05 |
| LMSM 8 | 275 | 880 | 0 |
| LMSM 12 | 24 | 110 | 4.95E-02 |
| LMSM 13 | 20 | 76 | 1.05E-02 |
| LMSM 14 | 252 | 1004 | 8.88E-16 |
| LMSM 16 | 48 | 182 | 1.11E-04 |

($p$-value < 0.05). In Table 2, 15 out of 17 LMSM modules are regarded as module biomarkers in BRCA at a significant level (Log-rank $p$-value < 0.05 and HR > 2). Particularly, 90% (9 out of 10, excepting LMSM 14) of the BRCA-related LMSM modules can act as module biomarker in BRCA. These results show that most of LMSM modules are functionally implicated in BRCA.

## LMSM modules are mostly BRCA subtype-specific

In this section, we firstly divide the 500 BRCA samples into five "intrinsic" subtypes (Luminal A, Luminal B, HER2-enriched, Basal-like and Normal-like). The numbers of LumA, LumB, Her2, Basal and Normal samples are 190, 155, 52, 85 and 18, respectively. Then we calculate the enrichment scores of the identified 17 LMSM modules in the BRCA subtype samples respectively (details in S5 Data).

As illustrated in Fig 3, out of the 17 LMSM modules, 4 and 6 modules are identified as up-regulated and down-regulated BRCA subtype-specific LMSM modules, respectively. For the up-regulated BRCA subtype-specific LMSM modules, the numbers of Basal-specific, LumB-

Table 2. **Survival analysis of LMSM modules in BRCA.** HRlow95 and HRup95 represent the lower and upper of 95% confidence interval of HR, respectively.

| Module ID | Chi-square | p-value | HR | HRlow95 | HRup95 |
|---|---|---|---|---|---|
| LMSM 1 | 170.37 | 0 | 10.75 | 5.88 | 19.65 |
| LMSM 2 | 107.34 | 0 | 6.03 | 3.12 | 11.66 |
| LMSM 3 | 90.62 | 0 | 5.43 | 2.94 | 10.01 |
| LMSM 4 | 138.81 | 0 | 14.94 | 8.83 | 25.27 |
| LMSM 5 | 148.49 | 0 | 8.64 | 4.63 | 16.13 |
| LMSM 6 | 142.64 | 0 | 13.40 | 7.83 | 22.92 |
| LMSM 7 | 161.91 | 0 | 13.97 | 8.01 | 24.36 |
| LMSM 8 | 103.63 | 0 | 5.91 | 3.07 | 11.37 |
| LMSM 10 | 144.86 | 0 | 8.63 | 4.74 | 15.71 |
| LMSM 11 | 120.79 | 0 | 9.49 | 5.55 | 16.23 |
| LMSM 12 | 49.31 | 2.19E-12 | 5.46 | 3.38 | 8.80 |
| LMSM 13 | 60.08 | 9.10E-15 | 5.72 | 3.48 | 9.41 |
| LMSM 15 | 83.26 | 0 | 12.00 | 7.46 | 19.32 |
| LMSM 16 | 110.94 | 0 | 11.25 | 6.79 | 18.66 |
| LMSM 17 | 106.96 | 0 | 9.14 | 5.42 | 15.41 |

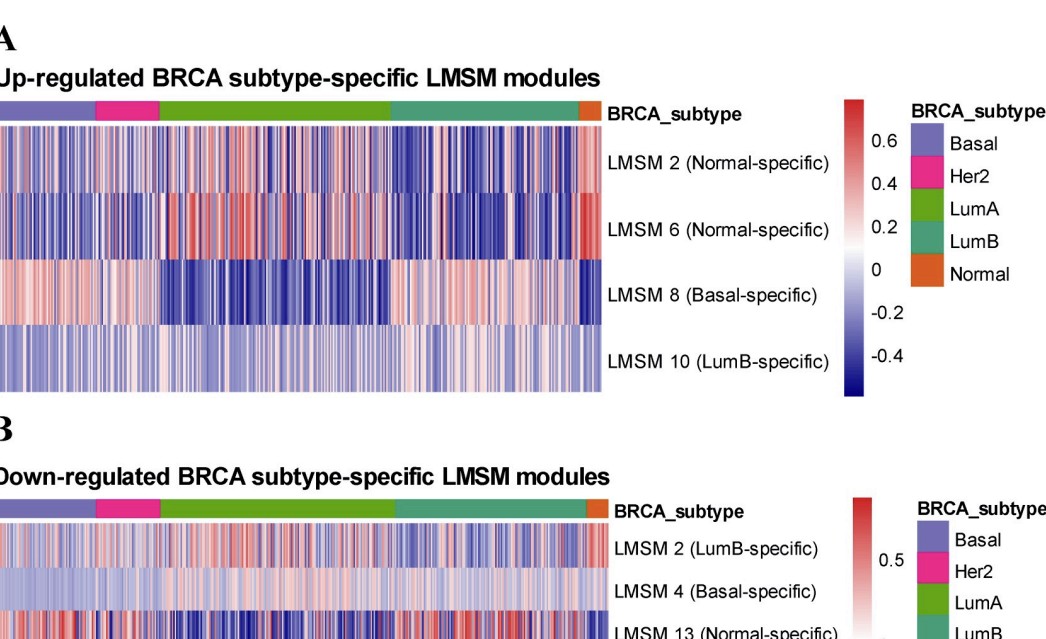

**Fig 3. Heatmap of the enrichment scores of BRCA subtype-specific LMSM modules in five BRCA subtype samples.** (A) Up-regulated BRCA subtype-specific LMSM modules. (B) Down-regulated BRCA subtype-specific LMSM modules.

specific and Normal-specific modules are 1, 1 and 2, respectively. The numbers of Basal-specific, LumB-specific and Normal-specific modules are 3, 1 and 2 respectively among the down-regulated BRCA subtype-specific LMSM modules. In particular, only 1 module (LMSM 2) can act as both up-regulated and down-regulated BRCA subtype-specific LMSM module. In total, the unique number of BRCA subtype-specific LMSM modules is 9, indicating that most of LMSM modules are BRCA subtype-specific.

## The performance of LMSM modules is significantly higher than baseline's performance in classifying BRCA subtypes

For the identified 17 LMSM modules, the average *Subset accuracy* and *Hamming loss* in classifying BRCA subtypes is 0.7547 and 0.0892, respectively (details can be seen in S6 Data), The *Subset accuracy* and *Hamming loss* of the baseline are 0.3800 and 0.2480, respectively. By using Welch's *t*-test method, the *Subset accuracy* achieved using the 17 LMSM modules is significantly larger (better) than the *Subset accuracy* of the baseline (*p*-value < 2.20E-16), and the *Hamming loss* of the 17 LMSM modules is significantly smaller (better) than the *Hamming loss* of the baseline (*p*-value < 2.20E-16). The better performance than the baseline method indicates that LMSM modules are biological meaningful in classifying BRCA subtypes.

## Several lncRNA-related miRNA sponge interactions are experimentally confirmed

For the ground truth used in the validation, we have collected 581 experimentally validated lncRNA-related miRNA sponge interactions associated with the matched lncRNA and mRNA

expression data (details in S4 Data). After we merge the sponge lncRNA-mRNA pairs in the identified 17 LMSM modules, we have predicted 1471664 unique lncRNA-related miRNA sponge interactions (details at https://github.com/zhangjunpeng411/LMSM). For each LMSM module, the number of shared miRNAs, lncRNAs, mRNAs, predicted lncRNA-related miRNA sponge interactions can be seen in S7 Data.

As shown in Table 3, there are 4 LMSM modules (LMSM 2, LMSM 3, LMSM 5 and LMSM 8) containing 14 experimentally validated lncRNA-related miRNA sponge interactions in total. It is noted that all the lncRNAs and mRNAs in these confirmed lncRNA-related miRNA sponge interactions are BRCA-related genes, indicating they may have potentially involved in BRCA.

## LMSM is capable of predicting miRNA targets

LMSM use high-confidence miRNA-target interactions as seeds to predict miRNA-target interactions. A miRNA-mRNA or miRNA-lncRNA pair in a LMSM module has the potential to be a miRNA-target pair for the following reasons. Firstly, at sequence level, the sponge lncRNAs and mRNAs in each LMSM module have a significant sharing of miRNAs. Secondly, at expression level, the sponge lncRNAs and mRNAs in each LMSM module are highly correlated. As a result, the sponge lncRNAs and mRNAs of each LMSM module have a high chance to be target genes of the shared miRNAs. Thus, based on the identified LMSM modules, we have predicted 2820524 unique miRNA-target interactions (including 2023304 miRNA-lncRNA and 797220 miRNA-mRNA interactions) (details at https://github.com/zhangjunpeng411/LMSM). For each LMSM module, the numbers of predicted miRNA-lncRNA interactions and miRNA-mRNA interactions can be seen in S7 Data.

In addition, we investigate the intersection of the miRNA-target interactions predicted by LMSM with the other well-cited miRNA-target prediction methods. In terms of miRNA-mRNA interactions, we select TargetScan v7.2 [51], DIANA-microT-CDS v5.0 [52], starBase v3.0 [53] and miRWalk v3.0 [54] for investigation. We choose starBase v3.0 [53] and DIANA-LncBase v2.0 [39] for investigation in terms of miRNA-lncRNA interactions. As shown in the UpSet plot [55] of Fig 4A, the number of miRNA-mRNA interactions identified by all the five methods is only 21842. However, the percentage of overlap between LMSM and each of the other four methods achieves ~63.74% (1289620 out of 2023304). As shown in Fig 4B, the number of miRNA-lncRNA interactions identified by all the three methods is only 1160. Since the miRNA-lncRNA interactions are still limited, most of the miRNA-lncRNA interactions (~93.90%, 748609 out of 797220) are individually predicted by LMSM.

## Comparison with graph clustering-based strategy

Graph clustering-based strategy [12–17] is an alternative approach to identifying lncRNA related miRNA sponge modules. As there is no graph clustering-based strategy specifically designed for finding lncRNA related miRNA sponge modules, so we create a baseline Graph

**Table 3. Validated lncRNA-related miRNA sponge interactions.**

| Module ID | Validated lncRNA-related miRNA sponge interactions |
|---|---|
| LMSM 2 | *H19: HMGA2, H19:IGF2, H19:ITGB1, H19: TGFB1, H19: VIM, H19:RUNX1, H19:CDH13, H19:KLF4, H19:TGFBI, H19:VDR* |
| LMSM 3 | *LINC00152: MCL1* |
| LMSM 5 | *NEAT1: EMP2* |
| LMSM 8 | *LINC00324: BTG2, DLEU2: CCNE1* |

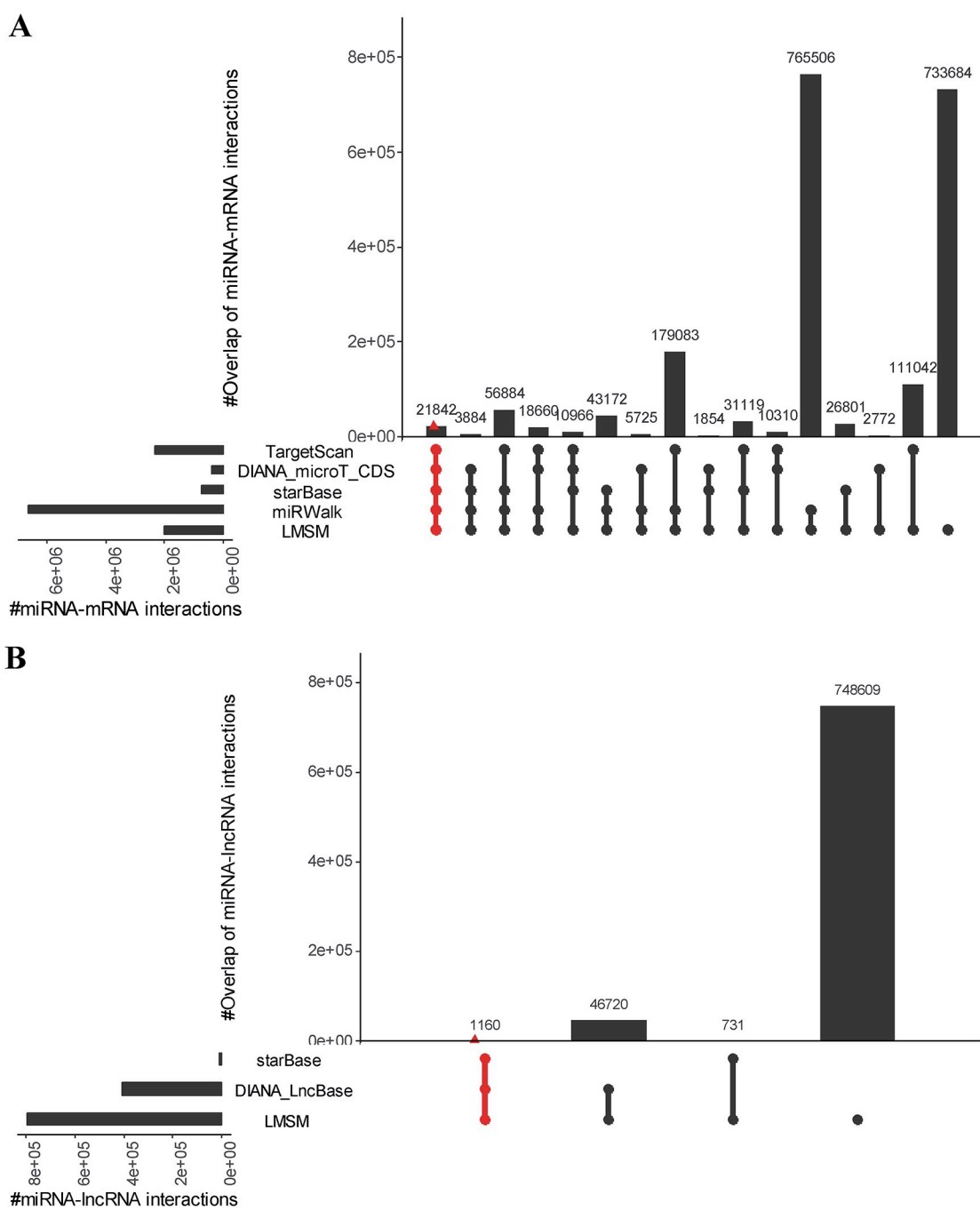

**Fig 4. Overlaps and differences between predicted miRNA-target interactions by LMSM and other methods.** (A) Predicted miRNA-mRNA interactions between LMSM and TargetScan, DIANA_microT_CDS, starBase, miRWalk. (B) Predicted miRNA-lncRNA interactions between LMSM and starBase, DIANA_LncBase. Each column corresponds to an exclusive intersection that includes the elements of the sets denoted by the dark or red circles, but not of the others. The overlap size between different methods denotes exclusive overlaps, i.e. the overlap set not in a subset of any other overlap set.

Clustering-based method (called GC in this paper) which uses well-known network construction and graph clustering methods as described in the following. The GC method includes two steps: i) identifying lncRNA related miRNA sponge interaction network, and ii) identifying lncRNA related miRNA sponge modules from the identified network. In step 1, we adapt the

**Table 4. Comparison results between LMSM and GC.**

| Method | %BRCA-related modules | %Module biomarkers | Mean *Subset accuracy* | Mean *Hamming loss* | #Validated interactions |
|---|---|---|---|---|---|
| LMSM | **58.82%** | **88.24%** | **0.7547** | **0.0892** | **14** |
| GC | 32.41% | 66.67% | 0.6586 | 0.1319 | 2 |

well-cited Sensitivity Correlation (SC) method [24] implemented in the *miRspongeR* R package [56] to infer lncRNA related miRNA sponge interaction network. A lncRNA-mRNA pair is considered as an interacting pair in the network if they have significant sharing of the miR-NAs, significant correlation and adequate sensitivity correlation. We require that the pairs must share at least 3 miRNAs and their sensitivity correlation (the difference between correlation and partial correlation) must be larger than 0.1. The statistically significance of the miRNA sharing and positive correlations are tested using hypergeometric test and Welch's *t*-test respectively, with a significant level at 0.05. In step 2, we use the well-cited Markov cluster (MCL) algorithm [57] to infer lncRNA related miRNA sponge modules. Here, each obtained cluster corresponds to a module. Each module should contain at least 2 sponge lncRNAs and 2 target mRNAs. In total, by using the GC method, we have obtained 108 lncRNA related miRNA sponge modules.

We compare LMSM and GC in terms of the percentage of BRCA-related modules, the percentage of module biomarkers in BRCA, the classification performance (mean *Subset accuracy* and mean *Hamming loss*) in classifying BRCA subtypes, and the number of validated lncRNA-related miRNA sponge interactions. As shown in Table 4, the comparison result indicates that LMSM always performs better than the GC method. The detailed results of the GC method can be seen in S8 Data.

## LMSM is robust

To demonstrate the robustness of the LMSM workflow, we use the sparse group factor analysis (SGFA) method [58], instead of the WGCNA method to identify lncRNA-mRNA co-expression modules. The SGFA method is extended from the group factor analysis (GFA) method [59–61], and it can reliably infer biclusters (modules) from multiple data sources, and provide predictive and interpretable structure existing in any subset of the data sources. Given *B* biclusters to be identified, the SGFA method assigns each column (lncRNA or mRNA) or row (sample) a grade of membership (association) belonging to these biclusters. The range of the values of the associations is [−1, 1]. We use the absolute value of association (*AVA*) to evaluate the strength of lncRNAs and mRNAs belonging to a bicluster, and the cutoff of *AVA* is also set to 0.8. Specifically, we use the *GFA* R package [58] to identify lncRNA-mRNA co-expression modules. The parameter settings for inferring lncRNA-related miRNA sponge modules are the same.

By using the SGFA method, we have identified 51 LMSM modules (details can be seen in S1 Data). The average size of these LMSM modules is 277.63 and the average number of the shared miRNAs is 135.65. There are 490 unique miRNAs mediating the 51 LMSM modules, and 84.90% (416 out of 490) miRNAs mediate at least two LMSM modules (details can be seen in S2 Data). As the result obtained using the WGCNA method, the result with the SGFA method also implies that the mediating miRNAs mostly act as crosslinks across different LMSM modules. In addition, by using a null-model-based *p*-value computation method, the identified 51 LMSM modules are also all statistically significant with adjusted *p*-value ≤ 5.00E-06 (details can be seen in S3 Data).

As shown in Table A of S1 File, 3 out of the 51 LMSM modules are functionally enriched in BRCA at a significant level (*p*-value < 0.05). Moreover, 49 out of the 51 LMSM modules are regarded as module biomarkers in BRCA (see in Table B of S1 File). The results indicate that most of LMSM modules are related to BRCA.

We also compute the enrichment scores of the identified 51 LMSM modules in the BRCA subtype samples (details in S5 Data). As illustrated in Fig A of S1 File, out of the 51 LMSM modules, 33 and 24 modules are regarded as up-regulated and down-regulated BRCA sub-type-specific LMSM modules, respectively. For the up-regulated BRCA subtype-specific LMSM modules, the numbers of Basal-specific, Her2-specific, LumB-specific and Normal-specific modules are 27, 2, 2 and 2, respectively. The numbers of Basal-specific, Her2-specific, LumA-specific, LumB-specific and Normal-specific modules are 2, 3, 15, 3 and 1 respectively for the down-regulated BRCA subtype-specific LMSM modules. Particularly, 16 modules can act as both up-regulated and down-regulated BRCA subtype-specific LMSM module. Overall, the unique number of BRCA subtype-specific LMSM modules is 41. This result also indicates that the identified LMSM modules are mostly BRCA subtype-specific.

The average value of *Subset accuracy* and *Hamming loss* of the identified 51 LMSM modules in classifying BRCA subtypes is 0.6921 and 0.1135, respectively (details can be seen in S6 Data). In classifying BRCA subtypes, the baseline value of *Subset accuracy* and *Hamming loss* is 0.3800 and 0.2480, respectively. By using Welch's *t*-test method, the value of *Subset accuracy* for 51 LMSM modules is significantly larger (better) than the baseline value of *Subset accuracy* (*p*-value < 2.20E-16), and the value of *Hamming loss* for 51 LMSM modules is significantly smaller (better) than the baseline value of *Hamming loss* (*p*-value < 2.20E-16). The better performance than the baseline method also indicates that LMSM modules are biological meaningful in classifying BRCA subtypes.

Moreover, we have predicted 605456 unique lncRNA-related miRNA sponge interactions in the identified 51 LMSM modules (details at https://github.com/zhangjunpeng411/LMSM). The number of the shared miRNAs, lncRNAs, mRNAs, predicted lncRNA-related miRNA sponge interactions of each LMSM module can be seen in S7 Data. Since the experimentally validated lncRNA-related miRNA sponge interactions are still limited, only 4 LMSM modules containing 4 lncRNA-related miRNA sponge interactions (see Table C of S1 File) are experimentally validated. All lncRNAs and mRNAs in the confirmed lncRNA-related miRNA sponge interactions are also BRCA-related genes.

LMSM also has identified a large number of potential miRNA-target interactions (1646449 in total, including 435345 miRNA-mRNA and 1211104 miRNA-lncRNA interactions, details at https://github.com/zhangjunpeng411/LMSM). The number of predicted miRNA-lncRNA interactions, predicted miRNA-mRNA interactions, putative miRNA-lncRNA interactions and putative miRNA-mRNA interactions can be seen in S7 Data. As illustrated in Fig B of S1 File, the number of the miRNA-mRNA interactions identified by all the five methods is 4897 and the number of the miRNA-lncRNA interactions identified by all the three methods is 1149. Most of the identified miRNA-mRNA interactions by LMSM (~58.55%, 254910 out of 435345) are also predicted by one of the other four methods. In terms of the predicted miRNA-lncRNA interactions, ~94.23% (1141232 out of 1211104) miRNA-lncRNA interactions are also individually predicted by LMSM.

Finally, in terms of the percentage of BRCA-related modules, the percentage of module biomarkers in BRCA, the classification performance (mean *Subset accuracy* and mean *Hamming loss*) in classifying BRCA subtypes, and the number of validated lncRNA-related miRNA sponge interactions, LMSM also generally performs better than the GC method (see Table D of S1 File).

Altogether, the above results are consistent with those obtained using the WGCNA method, indicating that our LMSM workflow is robust for studying lncRNA-related miRNA sponge modules.

## Discussion

The crosstalk between different RNA transcripts in a miRNA-dependent manner forms a complex miRNA sponge interaction network and depicts a novel layer of gene expression regulation. Until now, several types of RNA transcripts, e.g. lncRNAs, pseudogenes, circRNAs and mRNAs, have been confirmed to act as miRNA sponges. Since lncRNAs are a large class of ncRNAs and function in many aspects of cell biology, including human cancers, we focus on identifying lncRNA related miRNA sponge modules in this work.

By integrating multiple data sources, previous studies mainly investigate the identification of lncRNA related miRNA sponge interaction network. Based on the identified lncRNA related miRNA sponge interaction network, they use graph clustering algorithms to further infer lncRNA related miRNA sponge modules. Different from existing computational methods on lncRNA related miRNA sponge modules, in this work, we propose a novel method named LMSM to directly identify lncRNA related miRNA sponge modules from heterogeneous data. It is noted that the LMSM method depends on our presented hypothesis of miRNA sponge modular competition. In the hypothesis, miRNA sponges tend to form a group to compete with a group of target mRNAs for binding with miRNAs.

We have applied the LMSM method to the BRCA dataset from TCGA. For the putative miRNA-target interactions, we integrate high-confidence miRNA-target interactions from several databases. The analysis results demonstrate that our LMSM method is useful in identifying lncRNA related miRNA sponge modules, and it can help with understanding regulatory mechanism of lncRNAs.

LMSM is a flexible method to investigate miRNA sponge modules in human cancer. Firstly, any biclustering or clustering algorithm (e.g. the joint non-negative matrix factorization methods presented by Deng *et al.* [18] and Xiao *et al.* [19]) can be plugged in stage 1 of LMSM to identify lncRNA-mRNA co-expression modules. The only condition for using these algorithms is that they can be used to identify biclusters or clusters from high-dimensional expression data. Secondly, LMSM is a parametric model, and the parameter settings of LMSM can be replaced according to the practical requirements of researchers. For example, the threshold of the three metrics in stage 2 for identifying lncRNA related miRNA sponge modules can be looser or stricter. Thirdly, LMSM can also be extended to study other ncRNA (e.g. circRNA and pseudogene) related miRNA sponge modules. For instance, if we change the matched lncRNA expression data and the miRNA-lncRNA interactions to matched circRNA expression data and the miRNA-circRNA interactions respectively, the pipeline of LMSM is to identify circRNA related miRNA sponge modules.

It is noted that each LMSM module contains many sponge lncRNAs and mRNAs, so it is hard to experimentally validate such a module by follow-up wet-lab experiments. This is a common issue of existing computational methods, including LMSM. We suggest that biologists can select some sponge lncRNAs and mRNAs of interest in each LMSM module, and then validate the modular competition between the selected sponge lncRNAs and target mRNAs. We believe that LMSM is still useful in shortlisting high-confidence sponge lncRNAs and mRNAs for experimental validation. For example, previous study [62] has shown that lncRNA *MIR22HG* is functionally complementary to lncRNA *H19*. In the identified LMSM module no. 2 (LMSM 2), lncRNA *H19* is experimentally validated to compete with 10 target mRNAs (*HMGA2*, *IGF2*, *ITGB1*, *TGFB1*, *VIM*, *RUNX1*, *CDH13*, *KLF4*, *TGFBI* and *VDR*).

Thus, biologists can select 2 lncRNAs (*H19* and *MIR22HG*) and 10 target mRNAs (*HMGA2*, *IGF2*, *ITGB1*, *TGFB1*, *VIM*, *RUNX1*, *CDH13*, *KLF4*, *TGFBI* and *VDR*) in LMSM 2 to validate the modular competition between them.

Taken together, based on the hypothesis of miRNA sponge modular competition, we propose a new approach to identifying lncRNA related miRNA sponge modules by integrating expression data and miRNA-target binding information. Our method not only extends the ceRNA hypothesis, but also provides a novel way to investigate the biological functions and modular mechanism of lncRNAs in BRCA. We believe that our method can be also applied to other human cancer datasets assists in human cancer research.

## Supporting information

**S1 Data. The identified LMSM modules.**
(XLSX)

**S2 Data. The distribution of the shared miRNAs in LMSM modules.**
(XLSX)

**S3 Data. Statistically significant analysis results of LMSM modules.**
(XLSX)

**S4 Data. BRCA-related genes and experimentally validated lncRNA related miRNA sponge interactions.**
(XLSX)

**S5 Data. The enrichment scores of the identified LMSM modules in the BRCA subtype samples.**
(XLSX)

**S6 Data. Classification analysis results of LMSM modules in classifying BRCA subtypes.**
(XLSX)

**S7 Data. The number of shared miRNAs, lncRNAs, mRNAs, predicted interactions for each LMSM module.**
(XLSX)

**S8 Data. The results of a graph clustering-based strategy.**
(XLSX)

**S1 File. Supporting file.** Supplementary file.
(DOCX)

## Acknowledgments

We thank Prof. Guanwen Fang for the support from the Yunnan young and middle-aged academic and technical leaders reserve talent program and the Yunnan ten thousand talent program-young top-notch talent.

## Author Contributions

**Conceptualization:** Junpeng Zhang, Nini Rao, Thuc Duy Le.

**Data curation:** Junpeng Zhang, Nini Rao, Thuc Duy Le.

**Formal analysis:** Junpeng Zhang, Taosheng Xu.

**Funding acquisition:** Junpeng Zhang, Nini Rao, Thuc Duy Le.

**Investigation:** Junpeng Zhang, Taosheng Xu, Lin Liu, Nini Rao, Thuc Duy Le.

**Methodology:** Junpeng Zhang, Taosheng Xu, Lin Liu, Wu Zhang, Chunwen Zhao, Sijing Li, Jiuyong Li.

**Project administration:** Junpeng Zhang, Nini Rao, Thuc Duy Le.

**Resources:** Junpeng Zhang, Lin Liu, Jiuyong Li, Nini Rao, Thuc Duy Le.

**Software:** Junpeng Zhang.

**Supervision:** Nini Rao, Thuc Duy Le.

**Validation:** Lin Liu, Jiuyong Li.

**Visualization:** Wu Zhang, Chunwen Zhao, Sijing Li, Jiuyong Li.

**Writing – original draft:** Junpeng Zhang, Taosheng Xu.

**Writing – review & editing:** Junpeng Zhang, Taosheng Xu, Lin Liu, Nini Rao, Thuc Duy Le.

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
