## [Decision Letter · Decision Letter 0]

20 Jan 2020

Dear Prof. Zhang,

Thank you very much for submitting your manuscript "LMSM: a modular approach for identifying lncRNA related miRNA sponge modules in breast cancer" for consideration at PLOS Computational Biology.

As with all papers reviewed by the journal, your manuscript was reviewed by members of the editorial board and by several independent reviewers. In light of the reviews (below this email), we would like to invite the resubmission of a significantly-revised version that takes into account the reviewers' comments.

We cannot make any decision about publication until we have seen the revised manuscript and your response to the reviewers' comments. Your revised manuscript is also likely to be sent to reviewers for further evaluation.

Sincerely,

Teresa M. Przytycka

Associate Editor

PLOS Computational Biology

William Noble

Deputy Editor

PLOS Computational Biology

Reviewer's Responses to Questions

**Comments to the Authors:**

Reviewer #1: The paper describes a framework, LMSM, to identify LncRNA related MiRNA Sponge Modules from heterogeneous data. To understand the miRNA sponging activities in biological conditions, LMSM uses gene expression data to evaluate the influence of the shared miRNAs on the clustered sponge lncRNAs and mRNAs. This may be an important paper in the field however

1) I do not see any control experiments. There is some overlap with previous results however a statistical significance of this overlap is not clear at all. There are many lncRNAs, some overlap is expected due to random reasons.

2) There is no attempt to check biological relevance of the obtained module classification. I can not understand a quality of the classification. May be some experiments will improve the paper.

Reviewer #2: In their manuscript "LMSM: a modular approach for identifying lncRNA related miRNA sponge modules in breast cancer", Zhang et al. present a novel method for investigating the competing endogeneous RNA effect of long non-coding RNAs (lncRNA) on the level of modules. Here LMSM differs from existing methods that focus on pairwise consideratoins of lncRNA and mRNAs. The authors refer to this as miRNA sponge modular competition hypothesis. To obtain modules, LMSM uses the well established method WCGNA. Modules containing lncRNA as well as mRNAs (considered as two groups) are then evaluated using three criteria (i) significant number of shared miRNAs assessed using the hypergeometric test, (ii) canonical correlation analysis to assess the influence of the lncRNA group over the mRNA group, (iii) miRNA expression data is included to compute the partial-correlation based sensitivity correlation which was first introduced by Paci et. al. Sensitivity correlation is modified here for the use of partial canonical correlation to consider the effect of the lncRNA group to the mRNA group.

# Major:

- While it is appreciated that the authors released their source code on github, the software should also be documented, i.e. a README with installation instructions, dependencies and usage examples is needed.

- The authors differentiate between competing RNAs such as lncRNA or circRNAs and the mRNAs as a target group. I do not agree with this view as it neglects the fact that mRNAs act as competitors as well. For example, a highly expressed protein-coding mRNA may sponge miRNAs to increase the availability of other mRNAs coding for important interacting proteins, thus allowing the formation of protein complexes. This limitation should be discussed more critically in the manuscript.

- In the Methods, the authors state that "the available resources of lncRNAs are more abundant than those of other coding RNAs, circRNAs and pseudogenes". I do not understand this statement as clearly coding mRNAs are more abundant than lncRNAs.

- In WCGNA, R2 is empirically set as 0.8. Can you elaborate what you mean by empirically?

- In the third module assessment step, sensitivity correlation was used to quantify the sponging effect based on matched gene and miRNA expression data. We could recently show a major issue with sensitivity correlation is that it is biased by several factors including gene-gene correlation, sample number and number of shared miRNAs (List et al., https://doi.org/10.1093/bioinformatics/btz314). This bias can be adjusted for by using a suitable null model for infering significance. Could this null model approach described in the above paper be adapted towards the use partial canonical correlation?

- What is the source for the interactions of H19 and NEAT1? H19 and IGF2 for example share a complex relationship in imprinting that is not likely explainable by miRNA regulation. Thus, the results of module 2 should be seen critically.

- The main innovation of this method is that is infers miRNA sponge modules directly rather than first generating a comprehensive network which is then split into modules via graph clustering approaches. While this is indeed a promising strategy, the manuscript lacks a quantitative comparison to show advantages over the graph clustering strategy.

# Minor

- Table 1, column N2 is superfluous, N2 can be mentioned once in the caption.

- The interaction H19:TGFB1 is listed twice in Table 3.

**Have all data underlying the figures and results presented in the manuscript been provided?**

Reviewer #1: Yes

Reviewer #2: Yes

PLOS authors have the option to publish the peer review history of their article (what does this mean?). If published, this will include your full peer review and any attached files.

Reviewer #1: No

Reviewer #2: Yes: Markus List
---

## [Decision Letter · Decision Letter 1]

6 Apr 2020

Dear Prof. Zhang,

We are pleased to inform you that your manuscript 'LMSM: a modular approach for identifying lncRNA related miRNA sponge modules in breast cancer' has been provisionally accepted for publication in PLOS Computational Biology.

Best regards,

Teresa M. Przytycka

Associate Editor

PLOS Computational Biology

William Noble

Deputy Editor

PLOS Computational Biology

Reviewer's Responses to Questions

**Comments to the Authors:**

Reviewer #1: The authors tried to implement suggestions, the paper is acceptable.

Reviewer #2: All my previous concerns have been addressed adequately. The inclusion of a null model as well as a practical use case (breast cancer subtype classification) strongly increases confidence in the significance of the extracted modules. A comparison against the graph clustering approach is also highly appreciated and strengthens the manuscript considerably.

**Have all data underlying the figures and results presented in the manuscript been provided?**

Reviewer #1: Yes

Reviewer #2: Yes

PLOS authors have the option to publish the peer review history of their article (what does this mean?). If published, this will include your full peer review and any attached files.

Reviewer #1: No

Reviewer #2: Yes: Markus List

---

## [Editor Report · Acceptance letter]

10 Apr 2020

PCOMPBIOL-D-19-01970R1 

LMSM: a modular approach for identifying lncRNA related miRNA sponge modules in breast cancer

Dear Dr Zhang,

I am pleased to inform you that your manuscript has been formally accepted for publication in PLOS Computational Biology. Your manuscript is now with our production department and you will be notified of the publication date in due course.

With kind regards,

Sarah Hammond
